# Mapping Land-Use/Land-Cover Change in a Critical Biodiversity Area of South Africa

**DOI:** 10.3390/ijerph181910164

**Published:** 2021-09-27

**Authors:** Khangwelo Desmond Musetsho, Munyaradzi Chitakira, Willem Nel

**Affiliations:** Department of Environmental Sciences, College of Agriculture and Environmental Sciences, University of South Africa, Science Campus, Johannesburg 1710, South Africa; chitam1@unisa.ac.za (M.C.); willienel@wbs.co.za (W.N.)

**Keywords:** land-use/land-cover change, land-use planning, critical biodiversity area, traditional authority

## Abstract

Land-use/land-cover (LULC) changes have implications for the long-term outlook of environmental processes, especially in the face of factors such as climate change. These changes can have serious consequences for humans. In this study, remote sensing and geographic information system methods were used to investigate LULC changes in a critical biodiversity area (CBA) in the northern sections of Limpopo Province in South Africa from 1990 to 2018 using data obtained from the South African National Land Cover project. In 1990, the dominant land cover comprised thickets and dense bush, followed by woodland and built-up areas, covering proportions of 40, 24 and 18% of the total land-cover area, respectively. Bare and forest areas were the least dominant classes during this time. In 2018, the dominant land cover was woodland, followed by built-up areas, comprising 71 and 20% of the total area, respectively. Subsistence agriculture is a land-cover class with a relatively higher area compared to water bodies, wetlands and other classes. Between 1990 and 2018, significant changes in land-cover were noted for thickets and dense bush, woodland, water bodies, subsistence agriculture and built-up areas. Woodland increased by over 1000 hectares (ha) per year, while thickets decreased by over 900 ha per year. Interviews were conducted with local residents to determine what they thought were the drivers behind the observed changes. According to these interviews, the drivers included deforestation, agricultural activities in wetlands, sand and gravel mining, among others. The study’s outcomes are critical for future land-use planning exercises and the long-term conservation of this CBA, an area rich in biodiversity and a strategic water source for the communities.

## 1. Introduction

The world’s population is expected to increase to 10 billion by 2050, which is an increase of 2 billion people in 2019 [1]. This population explosion has implications for land-use/land-cover (LULC), as people continue to rely on the land for their survival when it comes to both food and shelter. The rate of LULC change has implications for the ecosystem services provided by the land/environment. Understanding the drivers behind LULC change is critical if the process of change is to be better understood. While this is crucial for all areas, the stakes are even higher for poor rural areas, some of which are critical biodiversity and ecological support areas. The management of land in rural areas of South Africa is in the spotlight, especially in various national debates about the need for more development activities in these areas [2]. The limited knowledge and understanding of the rate at which LULC change takes place, together with the associated implications for ecosystems and ecosystem services, is a cause for concern [3]. This concern highlights a need for better understanding of the current rate of LULC change and policy directions by those charged with managing natural capital through comprehensive research. Without this understanding, policy decisions related to land use and ecosystem services may be made that are not aligned with the practical realities on the ground and the anticipated rural economic development initiatives that have placed rural areas in the spotlight in South Africa.

The study setting is a critical biodiversity area (CBA) under the jurisdiction of a traditional leadership structure (the Mphaphuli Traditional Authority), which is considered rural and therefore a target for accelerated development, as alluded to by [4] in his maiden state-of-the-nation address. These accelerated development activities could be responsible for LULC changes. Having a scientific understanding of the rate of LULC changes is critical, and understanding the perceptions of land users and traditional leaders regarding what they think is behind such changes is also critically important in order to advise on how accelerated development activities could be undertaken sustainably. The rate of LULC changes could directly relate to how land is allocated for various uses. The way such use is regulated in a particular area and the associated policies, or lack thereof, around such activities are important. Therefore, understanding changes in LULC is critical for sustainable development [5].

Global and regional trends in land-cover changes have been established over time. Changes in land use take place at various spatial and temporal levels [6]. These changes have, at times, been beneficial, and at other times detrimental [7]. Anthropogenic factors are mainly responsible for affecting the structure and functioning of ecosystems (and, ultimately, the Earth system) and human well-being [8]. It has been noted that LULC changes are considered one of the main drivers that influence changes throughout the world. This view is supported by the assertions of other researchers, such as [9,10,11].

Other scholars such as [12] support the influence of land-use changes by arguing that they have serious implications for the world’s ecosystems, such as reducing the ability of these ecosystems to regulate climate change and provide food and shelter for both human beings and animals. Understanding how LULC changes affect the ability of ecosystems to provide value to landowners is an essential precondition for identifying efficient land-use patterns that maximise net social benefits [13]. Modification of ecosystems associated with critical biodiversity and ecological support areas takes place slowly in nature, but human-induced activities have been at the forefront of accelerated changes. The disturbances are poorly quantified and, as a result, their ecological states are poorly understood [14,15].

The objective of this study was to map land-use/land-cover (LULC) changes between 1990 and 2018 and, through interviews, to gather the perceptions of local community members regarding what they believe are the drivers of such changes. If the management of LULC changes is to be successful, it must be grounded in an understanding of the factors responsible for such changes. To this end, evidence of previous drivers and current and future factors should be considered [16]. Various parts of the African continent face significant pressure due to an intensified need for economic development. This now jeopardises the established protection of the considerable natural resources in these areas [17]. Therefore, concrete evidence of unsustainable LULC changes is critical, while mapping is also critical for proper planning around land-use changes and ecosystem services [18].

Two critical questions were the subject of this study:(a)What is the extent of LULC changes for the 28 years under review (1990 to 2018)?(b)What are the driving forces behind any changes in LULC?

Scholars such as [19,20,21,22,23] studied LULC changes extensively through remote sensing and GIS, with their results highlighting the rates of change over time. Scholars such as [24] also contributed to the discourse around land-cover classification and mapping, pointing out the changes taking place over time. However, there is very limited information based on investigating the combination of remote sensing data and community perceptions dealing with the rate of change in LULC and identifying drivers of such changes, particularly among rural based traditional leaders responsible for land management. This concept was the subject of this study.

## 2. Study Area

The study area forms part of a designated critical biodiversity area, a portion of the Vhembe Biosphere Reserve [25], situated in Thulamela Local Municipality of Vhembe District Municipality, in Limpopo Province of South Africa (Figure 1).

The area is under competing jurisdictional authority of the Thulamela Local Municipality and the Mphaphuli Traditional Authority. This area forms part of the Soutpansberg Mountains, a renowned area of endemism and a strategic water source for both surface and groundwater [25]. Kruger National Park forms the boundary to the east. The area has a very high unemployment rate, which is estimated at 44% [26], with almost 54% of households headed by women and an average household size of five people. Only 12% of the population has piped water inside their dwellings.

The area’s relief consists of undulating terrain with plains, hills and mountains that cover an area of approximately 67,000 hectares (ha). The climate is primarily influenced by the intertropical convergence zone [27]. Rainfall distribution in the area is classified as unimodal, with a rainy season predominantly between October and January [28]. The average annual rainfall is between 200 and 400 mm [29]. The northernmost part of the site is located at 22°37′10.97″ S and 30°45′54.07″ E, the southernmost part at 23°04′52.55″ S and 30°28′28.69″ E, the easternmost part at 23°04′00.98″ S and 30°27′05.50″ E, and the westernmost part at 22°38′39.04″ S and 30°56′20.08″ E.

## 3. Materials and Methods

The National Land Cover (NLC) products of South Africa between 1990 and 2018 were used as data sources [30]. The 1990 NLC data were derived from Landsat images at 30 m spatial resolution.

The overall accuracy of the 2018 NLC product is over 80%, but 1990 was not evaluated due to the paucity of relevant references. For the 2018 period, land-cover information was obtained from Sentinel-2 images at a spatial resolution of 20 m. These images were then resampled at 30 m for compatibility and comparability with those of the 1990 NLC.

Both land-cover maps had 72 classes, which were grouped into 11 classes based on the objectives of the study (Table 1; see land-cover map reports and legend at https://egis.environment.gov.za/, accessed on 20 May 2020). Since the land cover was surveyed using different satellite imagery (for 1990, the only available free-to-access medium resolution Landsat imagery was used and for 2018, the European Space Agency’s free-to-access Sentinel 2 satellite imagery was used), the legend was harmonised and standardised to enable comparison (Table 1). In the end, evidence for the study was gathered through geographical information system (GIS) mapping, ground truthing, and interviews with knowledgeable participants about the area under study.

After the land-cover maps and legends were harmonised, a postclassification method for land-cover change analysis was used in ArcGIS 10× (Figure 2), outlining a schematic representation of how the data analysis was implemented.

Postclassification is a simple method for determining land-cover changes that involves comparing the extent and areas of land-cover classes between two periods or points in time; it is also known as bitemporal change detection [19,20,21,22,23].

Postclassification provides the direction of change, i.e., from one point in time to another point. In this study, the years of comparison were 1990 to 2018. The change detection matrix and statistics were generated using a ha/year formula. The rate of change (ha/year) formula is described by [31] as follows:*R*Δ = (*t*2 − *t*1)/*z*(1)
where *R*Δ is the rate of change and *t* is the time of an area (ha) in years.

The amount of alteration in an area of land-cover class from an initial time (*t*1) to a later time (*t*2), *z*, is measured as the time interval between *t*1 and *t*2 in years. In this study, *z* was 28 years.

The data from the interviews and questionnaires were analysed with SPSS software. These data were mostly related to the perceptions of traditional leaders and their knowledge of the drivers of LULC changes in the area.

In this study, the sample consisted of 12 traditional leaders and 24 land-use rights holders (land users) purposely selected to represent different regions across the study area or certain privileged positions. Nonprobability purposive sampling was used to select and interview traditional leaders and government officials who were knowledgeable about the drivers of land-use/land-cover changes in the area and could articulate their experiences and perceptions.

In purposive sampling, participants are selected according to the researcher’s discretion, as determined by the extent to which the researcher is familiar with the environment and the participants’ reality [32,33]. Purposive sampling is described as a nonprobability sampling method that involves a conscious selection of participants for inclusion in a study, which is commonly used in qualitative research, based on the researcher’s judgement regarding the participants’ representative qualities or specialised knowledge of the phenomenon being studied [34]. For this study, the identified participants were traditional leaders responsible for managing the land, and government officials whose work included land use application processes in the area.

The eligibility of the participants was based on the following criteria: (a) all types of traditional leaders in the area who held a village leadership position, (b) all types of traditional leaders who formed part of the Mphaphuli Traditional Authority and (c) government officials whose work included land matters ranging from planning to policy and regulations.

Any potential participant who did not comply with the above criteria was not chosen to participate in this study. The semistructured, open-ended questionnaires in this study were used to gather information from the chosen participants. The questionnaires addressed the following issues/themes: (a) the demographic background of the participants, (b) perceptions of drivers of LULC change and (c) any other aspects of concern related to the use of land.

Permission to conduct the study was granted by the Mphaphuli Traditional Authority, Vhembe District Municipality, Thulamela Local Municipality and the Mphaphuli Development Trust. The participants were provided with comprehensive information regarding the study and their participation. Furthermore, to ensure that the participants were fully informed and able to provide informed consent, the full study title, the confidentiality clause and an explanation of whom to contact if they required any clarification were provided.

In this study, reliability was ensured by the centralisation of the main question around drivers of land-use/land-cover change among the traditional leaders. This ensured that there was consistency and no deviation from the originally stated objectives of the study. The preliminary findings and conclusions were also presented to those who participated in the interviews, and one-on-one discussions were held to allow them to agree or disagree with the findings.

## 4. Results and Discussion

For 1990, the dominant land cover was thickets and dense bush, followed by woodland and built-up areas, covering proportions of 40, 24 and 18% of the total area, respectively (Figure 3).

Bare and forest areas were the least dominant classes during this time. For 2018, the dominant land cover was woodland, followed by built-up areas, comprising 71 and 20% of the total area, respectively. Subsistence agriculture is a land-cover class with a relatively greater area than water bodies, wetland and other classes (Figure 4).

Between 1990 and 2018, significant changes in land cover were noted for the classes: thickets and dense bush, woodland, water bodies, subsistence agriculture and built-up areas (Table 2).

Woodland increased by over 1000 ha per year, which is a three-fold change from 1990 to 2018. Thickets and dense bush decreased by over 900 ha per year, a 38.79% decline between 1990 and 2018 (Table 2 summarises the rate of change). It appears that there is a relationship between the decline of thicket and dense bush and the increase in woodland. Ground truthing revealed that fuelwood related to charcoal and general cutting for selling purposes had been observed in the area, which contributed to transforming the thickets and dense bush into woodland. The development of a vast water body (Nandoni Dam) led to the loss of over 500 ha of thickets and dense bush, woodland and grassland. Built-up areas increased by 733 ha, or 26 ha per year, from 1990 to 2018, a 1.09% change during that period. Wetland areas declined by 25 ha, or almost 1 ha per year, a −0.04% change. Bare areas increased by 91 ha, or 3 ha per year, a 1.09% change. 

The observed 25-fold increase in water bodies from 24 hectares in 1990 to 679 hectares in 2018 could be attributed to a decline of 192 hectares of thickets/dense bush, 442 hectares of woodland, 22 hectares of grassland and at least four hectares of subsistence agriculture.

Woodland, built-up areas, water bodies and bare areas increased in hectares and percentage between 1990 and 2018. In contrast, thicket and dense bush, subsistence farming, commercial farming, forest plantation and wetlands declined in hectares and percentage. The indigenous forest land-cover class remained the same between 1990 and 2018. Table 3 summarises the identified drivers of change and the affected land-cover classes, where 1 = construction of a new dam, 2 = soil erosion, 3 = overpopulation/poverty, 4 = climate change, 5 = drought, 6 = agricultural activities, 7 = residential developments/urban encroachment, 8 = unplanned development projects, 9 = deforestation/logging/fuelwood cutting, 10 = mining (sand and gravel), 11 = overgrazing, 12 = fire, 13 = wetland destruction, 14 = drying of springs, 15 = road construction, 16 = business development activities and 17 = other municipal infrastructure works.

Table 4 presents postclassification results that show the changes observed between 1990 and 2018 using a “from–to” two-way table, and specifically the relationship between the decline and increase in particular land-cover classes.

Members of the community were selected to give their views or perceptions on the drivers of changes in land-use and land-cover in the area. The data originated from the study’s questionnaires and interviews and mainly reflect the perception of respondents who hold positions in the traditional leadership structure and government departments and are responsible for land use management on what could be behind the LULC changes. A number of possible drivers were presented to obtain the respondents’ views on their contribution to the changes observed between 1990 and 2018. These included drivers such as fire, drought, erosion, agricultural activities, climate change, deforestation activities, mining (sand and gravel), unplanned development, overgrazing and residential development.

Prioritisation of the drivers for the identified land-cover classes was based on the responses to the questions. The responses regarding the drivers of change were ranked according to the percentage of respondents who perceived a particular factor as important in the area.

### 4.1. Woodland, Thickets/Dense Bush and Indigenous Forest

The rate of change for these three land-cover classes based on GIS shows that woodland increased substantially, and thickets and dense bush declined almost equally. Indigenous forest remained the same. Respondents pointed out that deforestation, agricultural projects, unplanned development and mining (sand and gravel) were the drivers behind the changes. Woodland is generally transformed from thickets/dense bush due to changes in the canopy cover. However, the community still generally perceives these land-cover classes as forest in the area. Forests are critical ecosystems that provide an array of valuable services for human well-being. These ecosystems are known for some distinctive roles, ranging from their ability to sustain biodiversity to being habitats for various species of fauna and flora, as well as carbon sequestration and mitigating climate change. Like wetlands, forests also have a stabilising role in soil conservation, and stream flow function to prevent water runoff [35,36].

The study area is one of the most impoverished communities in the country, where most people are still very reliant on nature to provide services, including fuelwood for energy. Although this has been changing, since almost 87% of the population now has electricity [26], the level of poverty remains very high, and because of this the reliance on forests for fuelwood persists. Of late, there is a trend of cutting firewood for sale to upmarket areas, which continues unabated. This is one of the driving forces behind the high rate of conversion from forest to woodland. According to the locals, deforestation is the biggest threat to the forests/thickets in the area, followed by agricultural activities and mining on a small scale (sand and gravel).

The respondents identified deforestation as a significant factor that drives change in indigenous forests; 63.4% of the respondents perceived deforestation as a major driver of change (Figure 5).

Among the respondents, 21.4% identified agricultural projects and related activities as drivers of change in this critical biodiversity area. Agricultural projects are among the factors that compete with grasslands and forests for space, as indigenous forests are transformed for such projects.

Unplanned development, in many cases, is a result of poor governance, a lack of adequate land allocation management strategies and ineffective policies around enforcement of land use [37,38]. However, unplanned development can arise in an area due to other factors, such as overpopulation and urban expansion. In this study, 7.14% of the respondents identified unplanned development as a driver of change.

Mining activities have contributed to economic development in many parts of the world [39]. In addition, mining provides employment opportunities and contributes to infrastructure development, such as roads and access to water. However, it is also responsible for major environmental impacts, such as acid mine drainage, which leads to impacts on terrestrial and aquatic ecosystems [39]. In this study, 7.14% of the respondents identified mining (gravel and sand) as one of the drivers of change in the area.

### 4.2. Wetlands and Water Bodies

The mapping of LULC changes by GIS revealed that the surface area of water bodies increased between 1990 and 2018 as a result of the building in 2005 of a new dam by the Department of Water and Sanitation in South Africa. This change marked a positive trend in water body changes, which in many cases would otherwise be a downward trend.

A downward trend was identified through remote sensing, as wetlands were found to have declined by 25 ha, or almost 1 ha per year, over the period under investigation. This finding was in line with what the respondents observed and complained about regarding water bodies, rivers and wetlands (Figure 5).

In this study, the respondents identified wetlands and rivers as being among the most critical ecosystems affected by the observed drivers of change in the area. For example, the wetlands in Sambandou have been subjected to indiscriminate clearing, cultivation, and sand mining for some time. The Makwarela wetlands have been subjected to intense residential development inside their boundaries. The same goes for the Tshifudi and Tshaulu wetlands, which are being transformed into grazing and agricultural lands. Residential development is the most cited driver, raised by 41.2% of the respondents, followed by wetland destruction activities such as clearing and draining mentioned by 29.4% of the respondents. Poverty and overpopulation were cited by 11.8% of the respondents as driving forces.

Residential development can potentially undermine the integrity of sensitive ecosystem services. For example, some developments have been placed in buffer zones of rivers and wetlands. Such placement has a negative impact on these ecosystems, as problems with pollution, erosion, and habitat fragmentation are often the end result.

Wetland destruction as a driver of change was identified by 29.4% of the respondents. Thus, their views support a point raised by [40], who reported that many wetlands have been destroyed worldwide.

The respondents also identified grazing and the use of wetlands as cultivation land among the main drivers of changes in the area. Closely associated with the impact on wetlands is the drying up of springs in the study area. While 17.6% of the respondents identified this factor, the drying up of springs is an issue that could easily be related to other unplanned development activities mentioned earlier.

Poverty and overpopulation in sensitive areas were identified by 11.8% of the respondents as drivers of change around rivers and springs. Many poor rural communities rely heavily on wetlands and river ecosystem services for survival and to sustain their livelihoods, which could be why so many of them end up occupying parcels of land inside or close to sensitive ecosystems. The overreliance of poor rural communities on wetlands and rivers where no proper regulation is in place contributes to overutilisation.

The participants identified the drivers as well as additional variables behind the declining state of rivers and wetlands in the area. Deforestation was identified by 64.3% of the respondents as an additional driver of change in rivers, while agricultural projects were mentioned by 21.4% of respondents as a factor driving change. Further, 7.14% of respondents identified unplanned development and small-scale mining among the additional drivers of change in rivers.

The rivers are, according to the respondents, deteriorating at an alarming rate. Although a water quality analysis was not included in this study, one could infer that water quality could be a casualty of the deteriorating state of rivers in the area. Some villages in the CBA do not have an alternative source of water besides the local rivers and springs. These include communities such as Sambandou and Gunda, which have hardly any water flowing through installed standpipes.

Rivers in close proximity to these communities could suffer from pollution. Pollution in the CBA was evident during the fieldwork, as pollutants such as disposable diapers and litter were observed in and around the rivers. This emphasises the need for improvement in the management and monitoring of rivers to ensure that they remain free of pollution.

### 4.3. Grasslands

The respondents identified overgrazing and head cut soil erosion from disturbed grounds as significant drivers of change related to grasslands, followed by road construction, business development activities and other municipal infrastructure work. The respondents perceived four main factors as drivers of change around grasslands: overgrazing was identified by 35.7% as a major driver, road construction activities by 28.6% and general business development activities by 21.4%. Municipal infrastructure activities, in general, were identified by 14.3% of respondents as being behind changes in grassland ecosystems.

Grasslands are a major ecosystem, occupying one-third of the world’s terrestrial landscape, and are recognised globally for being rich in biodiversity [41]. However, grasslands are declining at an accelerated pace due to their conversion into arable land for agriculture [17]. The LULC mapping exercise revealed that 545 ha of grasslands were lost between 1990 and 2018.

Approximately 20% of grasslands elsewhere in the world have been irreversibly damaged due to their transformation to accommodate other land uses [42]. At the same time, grasslands have also been transformed into rangeland for livestock [43].

The health and productivity of global land resources such as grasslands are declining, while the demand for these resources increases [44]. Although they can support various forms of large stock units of game and livestock, grasslands are susceptible to overgrazing and soil erosion. The drivers of change are also area-specific, and areas with low rainfall and poor soil quality are more vulnerable to erosion and overgrazing. In contrast, grasslands with high rainfall and good soil quality are susceptible to leaching and the sourveld type of grass, which is more unpalatable for grazing.

Although roads are known for their role in connectivity and movement, they negatively affect a considerable amount of the landscape covered by grasslands. The impact of roads on grasslands is highly evident in areas such as Tshaulu, Tshifudi and Dimani, where grasslands once thrived.

### 4.4. Bare Land

It was found through GIS mapping that there was an increase in bare land across the study area. In 1990, there were 7 ha of bare land, and in 2018 this increased to 91 ha. The interviews showed that sand mining and deforestation are driving the change from land covered by either forest or woodland to bare land. Agricultural activities are also behind some of these changes. The mountainous areas of Tshidzini, Gaba, Tshifundi and Sambandou are most affected.

### 4.5. Built-Up Areas and Subsistence Farming

Built-up areas increased significantly between 1990 and 2018, closely linked to a decline in subsistence farming. Members of the community pointed out that land for subsistence agriculture has been preferred for extending built-up infrastructure. This is because the areas would have been transformed already, and are located much closer to other existing infrastructure. It was found by [45] that issues driving land-use studies include the removal or disturbance of productive land, urban encroachment and depletion of forests. The demand for residential sites due to increased population has been identified as a driver of change in this respect: from the 12,690 ha of built-up areas in 1990, 733 ha were added.

Land-cover changes due to urban expansion negatively impact biodiversity by causing degradation, loss of habitat and fragmentation [3,46], as highlighted by the respondents in this study. Such impacts have also been observed elsewhere around the world, such as in agriculture, siltation and concentrations of chemicals in the form of bioaccumulation, which affects the world’s threatened flora, fauna and other organisms [10].

## 5. Conclusions

Changes were observed in the land-cover classes from 1990 to 2018. The most significant change was for woodlands, which saw an increase from 16,299 hectares in 1990 to 47,906 hectares in 2018, a 46.94% increase. The second most significant change was the decline in thickets and dense forests, from 26,862 hectares in 1990 to 743 hectares in 2018, a 38.79% decline. A close relationship between woodlands and forests/thickets could be seen, as most of the hectares lost in the latter seem to have been gained by the former. Finally, land used for subsistence agriculture declined from 9713 hectares in 1990 to 3360 hectares in 2018, a 9.43% change. The natural process of change is anticipated because natural processes are dynamic; however, when anthropogenic factors interfere, changes happen at a much faster rate. The drivers behind the changes in LULC were identified through a rigorous process of interviews with local people, mostly government officials and traditional leaders. Through the interviews, we sought to solicit respondents’ views and perceptions of LULC changes.

The respondents were asked to identify and confirm the drivers behind the changes observed in the area, including fire, erosion, climate change, agricultural activities, deforestation, mining (sand and gravel), unplanned development, overgrazing and urban encroachment (residential development). Combining science with local knowledge contributes to a more holistic understanding of LULC changes and the associated drivers, as was the case in this study. There were significant LULC changes in this critical biodiversity area, which is also an ecological support area and a strategic source of both surface water and groundwater. There was a relationship between the decline in subsistence agricultural activities and the accelerated decline in thickets/dense forests. In addition, the decline of wetlands in a critical biodiversity area is cause for concern. Wetland areas are critical for the long-term availability of water for the local community and habitats for plant and animal species. It is thus crucial for changes in land-use/land-cover in this critical biodiversity area to be monitored closely and for interventions to be made at the policy level by those responsible for land management, i.e., the traditional authority and government officials. Educational or awareness efforts are necessary for the community members to learn the importance of the critical biodiversity area, and their sustainable management.

A detailed study of the links between land-cover classes that increased in hectares and those that declined is recommended in order to establish the real drivers of LULC change. Community perceptions may be motivated by site-specific encounters, whereas there may be many prevalent issues at a broader scale. 

## Figures and Tables

**Figure 1 ijerph-18-10164-f001:**
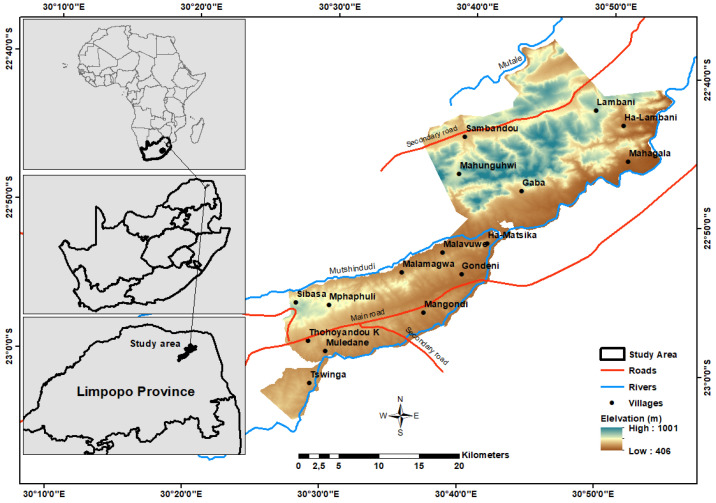
Location map of study area. (Source: digital elevation model, https://earthexplorer.usgs.gov/, accessed on 6 September 2021).

**Figure 2 ijerph-18-10164-f002:**
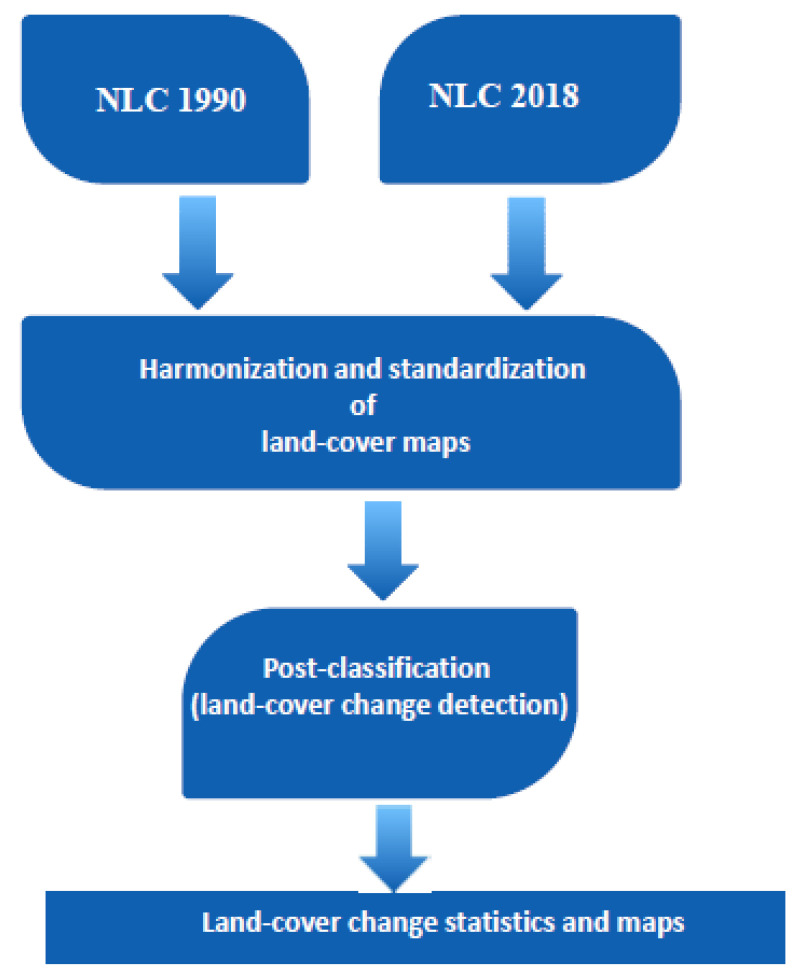
Schematic representation of land-cover change-analysis process; modified from [24].

**Figure 3 ijerph-18-10164-f003:**
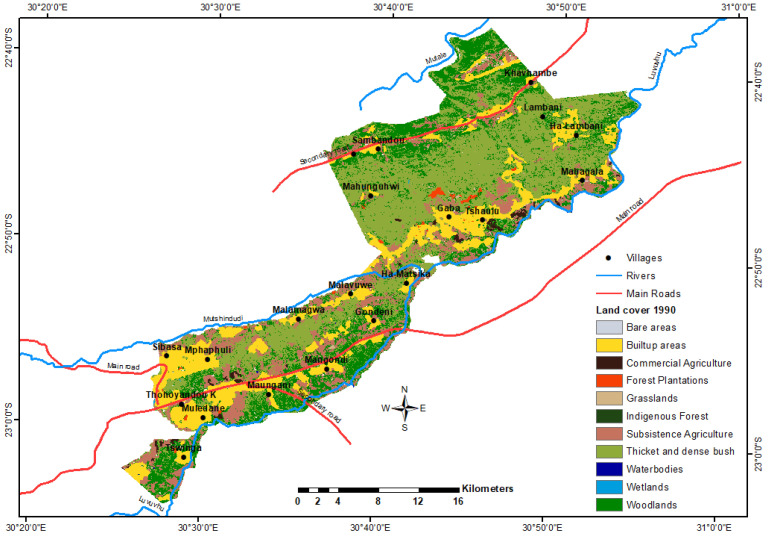
Study area land-cover map from South African National Land-Cover (SANLC) Project for 1990.

**Figure 4 ijerph-18-10164-f004:**
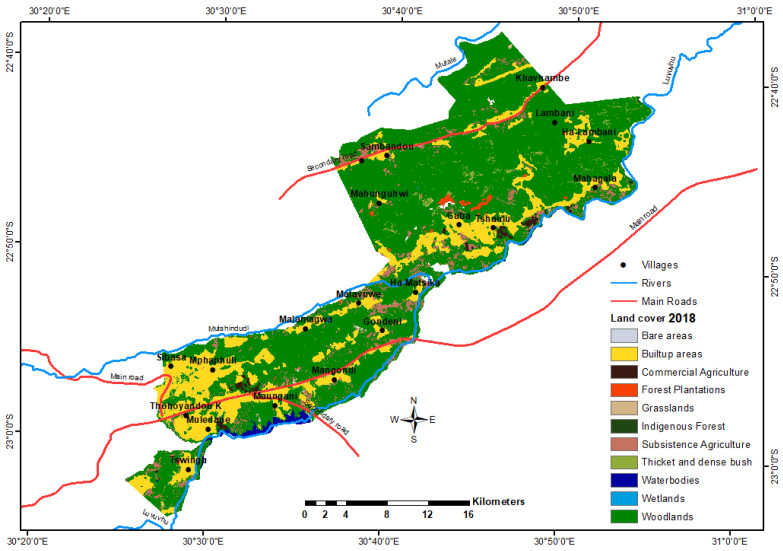
Study area land-cover map from SANLC Project for 2018.

**Figure 5 ijerph-18-10164-f005:**
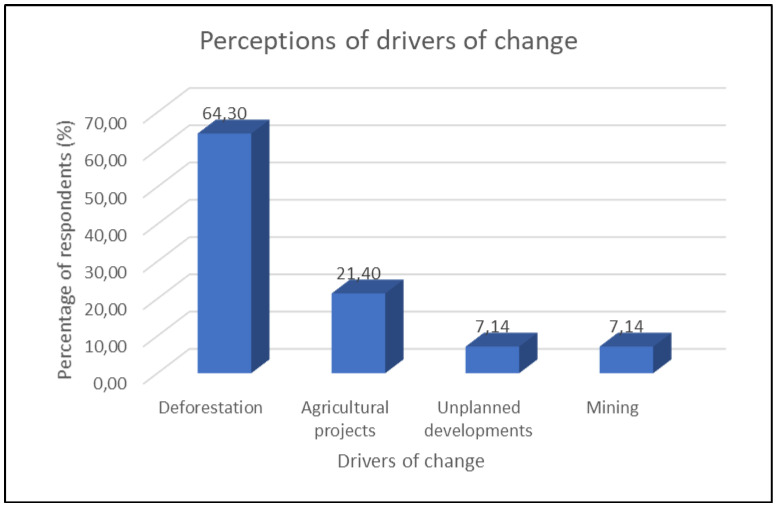
Perceptions of drivers of change in forests, wetlands and water bodies.

**Table 1 ijerph-18-10164-t001:** Standardised land-cover classes for changes between 1990 and 2018.

Land-Cover Type	New Classes	NLC 1990	NLC 2018
Water bodies	1	1–2	14–21
Wetlands	2	3	22–23, 73
Indigenous forest	3	4	1
Thicket and dense bush	4	5	2, 24
Woodland	5	6	3–4, 42–43
Grassland	6	7	12, 13, 44
Commercial agriculture	7	10–12, 26–31	32–40
Subsistence agriculture	8	23–25	41
Forest plantation	9	32–34	5–7
Bare areas	10	40–41	25–31, 45
Built-up areas	11	35–39, 42–72	47–72

Source: Researcher.

**Table 2 ijerph-18-10164-t002:** Area of each land-cover type in 1990 and 2018, and corresponding change analysis results.

Land-Cover Type	Value	1990 (ha)	1990 (%)	2018 (ha)	2018 (%)	Change (ha)	*R*Δ (ha/Year)	Ratio (%) Change
Water bodies	1	24	0.04	679	1.01	655	23	25.25
Wetlands	2	46	0.07	21	0.03	−25	−1	0.43
Indigenous forest	3	6	0.01	9	0.01	3	0	1.00
Thicket and dense bush	4	26,862	39.89	743	1.10	−26,119	−933	0.03
Woodland	5	16,299	24.21	47,906	71.15	31,607	1129	2.94
Grassland	6	801	1.19	256	0.38	−545	−19	0.32
Commercial agriculture	8	554	0.82	528	0.78	−26	−1	0.95
Subsistence agriculture	9	9713	14.42	3360	4.99	−6352	−227	0.5
Forest plantations	10	332	0.49	310	0.46	−22	−1	0.94
Bare areas	11	7	0.01	98	0.15	91	3	15.00
Built-up areas	12	12,690	18.85	13 423	19.94	733	26	1.06
		67,333	100	67,333	100			

*R*Δ: rate of change (ha/year).

**Table 3 ijerph-18-10164-t003:** A summary of the drivers of change and the affected land-cover classes.

		Identified Drivers of Change																
**Affected Land Cover** **Classes**		**1**	**2**	**3**	**4**	**5**	**6**	**7**	**8**	**9**	**10**	**11**	**12**	**13**	**14**	**15**	**16**	**17**
	Water bodies	X						X						X	X			
	Wetlands			X			X	X			X	X		X	X			
	Indigenous forest						X		X	X	X							
	Thicket and dense bush						X		X	X	X							
	Woodlands						X		X	X	X							
	Grasslands											X				X	X	X
	Commercial agriculture																	
	Subsistence agriculture			X			X			X	X							
	Forest plantation																	
	Bare areas						X			X	X							
	Built-up areas			X			X	X		X	X							

**Table 4 ijerph-18-10164-t004:** Postclassification results of changes in a two-way “from–to” format.

		NLC 2018											
	Land-Cover Type	1	2	3	4	5	6	7	8	9	10	11	Total Area (ha)
**NLC 1990**	1. Water bodies	8	0	0	0	14	0	0	0	0	1	0	24
	2. Wetlands	10	2	0	3	27	0	0	0	0	0	4	46
	3. Indigenous forest	0	0	0	3	3	0	0	0	0	0	0	6
	4. Thicket and dense bush	192	8	5	669	24,613	62	25	427	28	26	806	26,862
	5. Woodlands	442	3	4	30	14,445	130	27	228	45	55	891	16,299
	6. Grassland	22	0	1	2	635	17	3	19	11	7	84	801
	7. Commercial agriculture	0	0	0	4	171	0	342	33	0	0	3	554
	8. Subsistence agriculture	4	7	0	27	6947	12	116	2488	2	5	105	9713
	9. Forest plantations	0	0	0	1	56	2	1	4	224	0	45	332
	10. Bare areas	0	0	0	0	5	1	0	0	0	0	2	7
	11. Built-up areas	0	0	0	3	991	32	15	161	0	3	11,485	12,690
	Total area (ha)	679	21	9	743	47,906	256	528	3360	310	98	13,423	67,333

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
