# Peer review of "Mapping Land-Use/Land-Cover Change in a Critical Biodiversity Area of South Africa"

_ijerph, 2021, doi:10.3390/ijerph181910164_

Round 1

Reviewer 1 Report

  • Fig.1: The left side of the figure is OK, but location map should contain different information than LULC (it duplicates fig. 3). Shaded relief, (or coloured DEM) and other features of topography must be added.  
  • Line 81: I miss the detailed description of the NLC product. What method of LULC extraction was used? It is crucial for reliability and accuracy. What were the main errors of NLC 2018 with overall acurracy only 80%? Can this error rate affect the results of the 1990 and 2018 comparison?
  • fig 3 and fig 4 have different map projection (it can be seen in the shape of studied area), please correct it. 
  • fig 5 should demonstrate relative values. (e.g. 1990 as 100%). Changes in classes with low percentage of studied area are not visible even if they are very important (from 46 ha to 21 ha of wetlands - it is less than one half; bare areas (7 to 91) it is 13x more! Absolute values are in table 2 and it is sufficient.
  • chapter 4.1 I miss the overview (as a table) of all the drivers of change and what classes they affect. 
  • line 225 what is the difference between forest and woodland? How is it possible that the main forest change driver is deforestation (fig 6) even if the area of woodland increased substantially ?
  • lines 273-299 contain repetitive information, this section should be abbreviated.
  • lines 300-306: the drivers of change of forests really were identified with the exactly same values as the wetlands? 
  • chapter 4.1.3.
    • 329-333: The assertions are related to the area of interest or are they just general claims? (the same in lines 360-363). 
    • table 4 does not have any relation to the studied area. It shloud be either removed or relevance connection with the result must be added.
  • Any discussion of the results? 

Author Response

RESPONSE TO REVIEWER 1 COMMENTS

Point 1: Fig.1: The left side of the figure is OK, but location map should contain different information than LULC (it duplicates fig. 3). Shaded relief, (or coloured DEM) and other features of topography must be added.

Response 1:

An updated Figure 1 is now attached, with LULC information removed. The figure shows elevation, roads, rivers and villages.

Point 2: Line 81: I miss the detailed description of the NLC product. What method of LULC extraction was used? It is crucial for reliability and accuracy. What were the main errors of NLC 2018 with overall accuracy only 80%? Can this error rate affect the results of the 1990 and 2018 comparison?

Response 2:

Existing datasets were used as explained under materials and methods, there isn’t much that the authors could do about it.  Description of the NLC comes from the reports that could be found at https://egis.environment.gov.za, as explained under materials and methods. If those reports were to be included as part of the article, the article would be too long.  

Point 3: fig 3 and fig 4 have different map projection (it can be seen in the shape of studied area), please correct it.

Response. The authors used the Albers equal area projection for both figures.

Fig 5 should demonstrate relative values. (e.g. 1990 as 100%). Changes in classes with low percentage of studied area are not visible even if they are very important (from 46 ha to 21 ha of wetlands - it is less than one half; bare areas (7 to 91) it is 13x more! Absolute values are in table 2 and it is sufficient.

Response 3:

Percentage change was added in Table 2 and Figure 5 was deleted.

Point 4: chapter 4.1 I miss the overview (as a table) of all the drivers of change and what classes they affect.

Response 4:

Table 3 has been added as suggested, it summarises the drivers of change and the land-cover classes they affect.  

Point 5: line 225 what is the difference between forest and woodland?

How is it possible that the main forest change driver is deforestation (fig 6) even if the area of woodland increased substantially?

Response 5:

The land cover classes were obtained from the National Land Cover project (https://egis.environment.gov.za), which differentiates between Forest and Woodlands. The Authors did not change this description. There is a difference between forest and woodland – mainly due to canopy cover (forests are much closed whereas woodlands are open). The NLC classes are explained in the NLC project as per the url given. Woodland increased due to the decline of thicket/dense bush.

Point 6: lines 273-299 contain repetitive information, this section should be abbreviated.

Lines 300-306: the drivers of change of forests really were identified with the exactly same values as the wetlands?

Response 6:

Repetitive information deleted. Yes, same values for forest and wetlands were found.

Point 7: chapter 4.1.3.329-333: The assertions are related to the area of interest or are they just general claims? (the same in lines 360-363).

table 4 does not have any relation to the studied area. It should be either removed or relevance connection with the result must be added.

Any discussion of the results?

Response 7:

Changes were made to the chapter as suggested. Discussion of the results added to the article.

Reviewer 2 Report

This study investigated the LULC change in the northern parts of the Limpopo province in South Africa from 1990 to 2018. The drivers of such changes have also been investigated through interviews with local people. In my view, the novelty of this whole manuscript is limited. The only interesting part is to use interviews to understand drivers. But, the results and analyses seem to be subjective, and the effectiveness of using such an approach has not been verified. My comments are listed as follows.

  1. Page 1, L28-29: It is odd to start the manuscript from "objective" rather than background. Normally, we start from background, limitation(s) and problem(s), and then objective(s), in order to let readers understand the research questions and objectives in a paper.

  1. Page 1-2, Section 1. The whole section introduces the significance of why need to investigate LULC change. However, I believe that the issue has been well known for most readers. More important, I cannot see any comment or statement on current research limitation(s) or question(s) on investigating LULC change. Without limitations or questions, we cannot understand the objective and significance of this study.

  1. Page 3-4, ''Materials and Methods'': From lines 78 to 118, I cannot see any novel data or method used on investigating LULC change. So what is the contribution of this study, except that the authors used a different study area?

  1. Page 4, Line 119-127: The authors used questionnaires to investigate the drivers of LULC change. This may be the only part that interested me. However, the authors did not present how to design such questionnaires? How to verify such questionnaires are effective for investigating the drivers of LULC change? Without validation, we cannot simply believe that the results of questionnaires will reflect the truth. Moreover, only a few local people (36) have been asked to complete the questionnaires, it is also doubt that whether such a small sample size is able to understand the drivers of LULC change for such a large study area?

  1. Figure 6: only one figure was given out to analyze the drivers of LULC change, i.e., for the LULC type-Forests and woodlands. But no figure has been given out for other LULC types (e.g. wetland and water bodies) . What is/are the reason(s)?

Moreover, for some LULC types (e.g., bare lands and built-up areas), the authors even did not show the results of questionnaires. For instance, how much percentage of various different divers for these LULC types?

  1. Generally, the analyses in Section 4.1 seem to be subjective. I suggest the authors to use some quantitative method (e.g., regression analysis) as supplement for these questionnaires. Besides, I cannot find the section 4.2. However, it is not common to design a section called 4.1, but without following the section 4.2.

  1. The authors stated that ''the findings of this study should be the starting point towards the development of a land-use planning policy position for the area.'' Based on the analyzed divers, it is still not clear how can they be used for policy implications. I think this point should be discussed in more details.

Author Response

RESPONSE TO REVIEWER 2 COMMENTS

Point 1:

Page 1, L28-29: It is odd to start the manuscript from "objective" rather than background. Normally, we start from background, limitation (s) and problem(s), and then objective(s), in order to let readers understand the research questions and objectives in a paper.

Response 1:

Objectives were removed from the commencement of introductions to the end. Additional information was added to amplify the problems and objectives.

Point 2:

Page 1-2, Section 1. The whole section introduces the significance of why need to investigate LULC change. However, I believe that the issue has been well known for most readers. More important, I cannot see any comment or statement on current research limitation(s) or question(s) on investigating LULC change. Without limitations or questions, we cannot understand the objective and significance of this study.

Response 2:

Whilst we recognise the fact that the issue of LULC has been known for most readers, information has now been added to indicate the addition of an extra layer of investigation, that talks to the perceptions of traditional leaders and government employees responsible for land management. The study was limited to local scale, land under the jurisdiction of traditional authority. The authors do not claim to be developing any new methods of LULC investigations as such.

Point 3:

Page 3-4, ''Materials and Methods'': From lines 78 to 118, I cannot see any novel data or method used on investigating LULC change. So what is the contribution of this study, except that the authors used a different study area?

Response 3:

See response to point 2 above. The study was not about developing new methods, but to use conventional method to assist in determining LULC changes at local scale. This to link with local perceptions etc, incorporating these into land use planning.

Point 4:

Page 4, Line 119-127: The authors used questionnaires to investigate the drivers of LULC change. This may be the only part that interested me. However, the authors did not present how to design such questionnaires? How to verify such questionnaires are effective for investigating the drivers of LULC change? Without validation, we cannot simply believe that the results of questionnaires will reflect the truth. Moreover, only a few local people (36) have been asked to complete the questionnaires, it is also doubt that whether such a small sample size is able to understand the drivers of LULC change for such a large study area?

Response 4:

Information was added under materials and methods. The study was part of a bigger research activity that was undertaken, which looked at LULC, Ecosystem Services Valuation, Predicting future LULC for the area. Detailed explanation on the questionnaires has now been provided, including what they had to achieve. Purposive sampling was also explained in detail.

Point 5:

Figure 6: only one figure was given out to analyze the drivers of LULC change, i.e., for the LULC type-Forests and woodlands. But no figure has been given out for other LULC types (e.g. wetland and water bodies) . What is/are the reason(s)?

Moreover, for some LULC types (e.g., bare lands and built-up areas), the authors even did not show the results of questionnaires. For instance, how much percentage of various different drivers for these LULC types?

Response 5:

A lot of the information did not go into the journal due to limitations on the amount of information that could be added. That not withstanding, additional information on the results has been added, including on Table 3 and Figure 6.  

Point 6:

Generally, the analyses in Section 4.1 seem to be subjective. I suggest the authors to use some quantitative method (e.g.,regression analysis) as supplement for these questionnaires.

Besides, I cannot find the section 4.2. However, it is not common to design a section called 4.1, but without following the section 4.2.

Response 6:

We only explained the results from the questionnaires against the results of the GIS analysis. Linkages were established through a now detailed analysis. Regression analysis was not undertaken, but carried into the conclusion as a form of recommendation as this is an ongoing study.

Point 7:

The authors stated that ''the findings of this study should be the starting point towards the development of a land-use planning policy position for the area.'' Based on the analyzed drivers, it is still not clear how can they be used for policy implications. I think this point should be discussed in more details.

Response 7:

The statement on policy has been removed, but further details added for conclusion purposes, related to the findings of the study.

Reviewer 3 Report

The manuscript is about using Arc GIS to map proportion of different land use types in two years (1990 and 2018) and mapping the changes over a period of 28 years. Drivers of the changes were listed based on the responses from the interviews conducted with leaders and land users.  It is a simple study, with great potential to quantify major changes in the area and drivers that were identified by local stakeholders. However, the manuscript at this stage is poorly written. Introduction/background & discussion needs to be re-written. There is a huge lack in stating the background of the study, research questions, rationale, and objectives. Discussion is poorly written. Most of the information in discussion section need to be moved to introduction, supporting why this study is done. The study is simple and well documented to show the changes in land use, but more work needs to be done on the second part of the analysis which is identifying drivers of the changes. One way to improve presenting the results can be by identify the top 5 changes in the area and making multiple pi-charts identify the major drivers of the change. You have just one figure (fig. 6) showing drivers of forest change.

Instead of stating the area of change in ha, it would be nice to look at the proportion of change in each land use categories.  

Few editorial changes are:

Line 28: why is the paper started with objective? Please move the objective after introduction

Line 100: “was used” and “was implemented”, you can use wither of the two phrases

Line 111: confusing sentence

Line 124: Please add few citations for “Non-probability purposive sampling”

Line 155: Table 3 does not show the “drivers or causes of observed change”. Table 3 simply looks like change table (similar to accuracy assessment table) showing land use change from 1990 to 2018.

Line 174-186: It looks like background for this study not discussion.

Line 189-190: what were the drovers that were omitted. How did you decide their implications were negligible? Also, what do you mean by rate of change for drivers. Rate of change id for land use not drivers. What you have in Figure 6 are the “actual drivers”

Line 247: Seven percent is not a small scale.

Line 258-266: Not relevant at this part of discussion. This paragraph looks like part of introductory sections.

Line 267-268: Weird sentence – “the respondents …………….activities”.

Line 277-281: Repeated information, already stated in the earlier paragraph.

Line 291: 17% is a huge proportion, please omit saying “only” 17.6

Line 318: I don’t understand how soil erosion is a “driver” for grassland change. I would think soil erosion is the effect of grassland change.

Line 330: Is the 20% change in the study area?

Line 330-342: Unnecessary information in discussion. Table 4 is irrelevant. These are all general introduction on why grassland study is important. I do not see the relevance of this at this point of the manuscript.

Line 344-346: Add some quantitative numbers to support this statement.

Line 352-354: You need to have supportive data to confirm what is seen through the glance.

Line 366: what is “marked changes”

Line 370-372: This study does not identify the drivers using Arc GIS. Arc GIS is used only to measure the changes. Drivers are identified by interviews.

Line 375-379: The manuscript does not discuss about local solutions. Please avoid using new information in conclusion. Conclusion is supposed to summarize the major things of the study.

Author Response

RESPONSE TO REVIEWER 3 COMMENTS

Point 1:

The manuscript is about using Arc GIS to map proportion of different land use types in two years (1990 and 2018) and mapping the changes over a period of 28 years. Drivers of the changes were listed based on the responses from the interviews conducted with leaders and land users. It is a simple study, with great potential to quantify major changes in the area and drivers that were identified by local stakeholders. However, the manuscript at this stage is poorly written. Introduction/background & discussion needs to be re-written. There is a huge lack in stating the background of the study, research questions, rationale, and objectives. Discussion is poorly written. Most of the information in discussion section need to be moved to introduction, supporting why this study is done. The study is simple and well documented to show the changes in land use, but more work needs to be done on the second part of the analysis which is identifying drivers of the changes. One way to improve presenting the results can be by identify the top 5 changes in the area and making multiple pi-charts identify the major drivers of the change. You have just one figure (fig. 6) showing drivers of forest change.

Response 1:

We agree with the reviewer. Intro/discussions rewritten/revised, adding the human component (questionnaires, how they were designed etc). See detailed objectives and the added information in introduction and materials/methods

Point 2:

Instead of stating the area of change in ha, it would be nice to look at the proportion of change in each land use categories.

Response 2:

We agree, that information has been included in table 2 in percentages.

Few editorial changes are:

Point 3:

Line 28: why is the paper started with objective? Please move the objective after introduction

Response 3:

This has been addressed. Objectives moved towards the end of the introductory section.

Point 4:

Line 100: “was used” and “was implemented”, you can use either of the two phrases

Response 4:

Thank you for picking this up, the authors revised by deleting “Was implemented” as suggested.

Line 111: confusing sentence

Sentence revised.

Point 5:

Line 124: Please add few citations for “Non-probability purposive sampling”

Response 5:

A few citations were added around this non probability purposive sampling as suggested.

Point 6:

Line 155: Table 3 does not show the “drivers or causes of observed change”. Table 3 simply looks like change table (similar to accuracy assessment table) showing land use change from 1990 to 2018.

Response 6:

A new Table 3 has been added, with changes made to Table 4 as well. Table 4 remains in the manuscript as more discussions were based on it. The authors added more analysis of the drivers of change in the manuscript, associating the GIS findings with perceptions from the participants.

Point 7:

Line 174-186: It looks like background for this study not discussion.

Response 7:

The authors agree with the reviewer. The paragraphs has been moved to introductions.

Point 8:

Line 189-190: what were the drivers that were omitted. How did you decide their implications were negligible? Also, what do you mean by rate of change for drivers. Rate of change is for land use not drivers. What you have in Figure 6 are the “actual drivers”

Response 8:

Drivers of change which were the basis of the questionnaires have now been listed under results and discussions. This has also been summarised in Table 3.

Point 9:

Line 247: Seven percent is not a small scale.

Response 9:

Sentence rephrased.

Point 10:

Line 258-266: Not relevant at this part of discussion. This paragraph looks like part of introductory sections.

Response 10:

Section moved to introduction, where it was also further refined.

Point 11:

Line 267-268: Weird sentence – “the respondents…………….activities”.

Response 11:

Sentence has been revised, the authors agree with the reviewer.

Point 12:

Line 277-281: Repeated information, already stated in the earlier paragraph.

Response 12:

Information deleted.

Point 13:

Line 291: 17% is a huge proportion, please omit saying “only”17.6

Response 13:

The word “only was deleted”

Point 14:

Line 318: I don’t understand how soil erosion is a “driver” for grassland change. I would think soil erosion is the effect of grassland change.

Response 14:

Sentence revised to indicate that it is in fact head cut erosion from disturbed grounds that affects grasslands.

Point 15

Line 330: Is the 20% change in the study area?

Response 15:

Statement revised to indicate elsewhere, not the study area, this was part of the discussions.

Point 16:

Line 330-342: Unnecessary information in discussion. Table 4 is irrelevant. These are all general introduction on why grassland study is important. I do not see the relevance of this at this point of the manuscript.

Response 16:

The authors agree with the reviewer, Table 4 has now been deleted and the other paragraphs changed to amplify the discussions around the study area.

Point 17:

Line 344-346: Add some quantitative numbers to support this statement.

Response 17:

Discussions of the results now incorporate numbers.

Point 18:

Line 352-354: You need to have supportive data to confirm what is seen through the glance.

Response 18:

Statement revised.

Point 19:

Line 366: what is “marked changes”

Response 19:

We agree that the statement may be misleading, “marked” has been deleted and the paragraph rephrased.

Point 20:

Line 370-372: This study does not identify the drivers using ArcGIS. Arc GIS is used only to measure the changes. Drivers are identified by interviews.

Response 20:

The authors agree with the reviewer. Sentence has been amended

Point 21:

Line 375-379: The manuscript does not discuss about local solutions. Please avoid using new information in conclusion. Conclusion is supposed to summarize the major things of the study.

Response 21:

Paragraph deleted and information specific to the study included as part of the conclusion.

Reviewer 4 Report

This manuscript deals with Landuse Landcover mapping in the Critical Biodiversity area, South Africa. I could appreciate the authors for their work and interest. However, the manuscript needs to improve more. The comments and suggestions are as follows:

 L 27-58: Introduction part needs to add more recent reviews with significance. Author should add more details about study area such as Geographical coordinates, altitude, and temperature. There is no detailed methodology about data sources, collection and analysis, authors are requested to add in detail.

L 242-245: “Mining activities have contributed  to  economic  development  in  many  parts  of  the world (Clemens et al. 2018). Mining projects contribute to employment opportunities and infrastructure development, such as roads and access to water ecosystems …”. It’s a general statement. Mention the kind of mining activities in the study area and discuss more with your results.

L 362: “Some of  the  drivers of  change  include  agriculture,  siltation,  and  concentration  of chemicals….. which affects the world’s threatened flora,  fauna, and organisms…”. Is it same drivers in your study area?  Please discuss with your results.

  • Table and Figures: Should correct the text formats uniformly (Eg: Legend; Axis Title)
  • Add coordinates in all the figures;

Whole manuscript needs to format according to the journal instructions. I recommend the manuscript for publication with “Minor Revision”

Author Response

RESPONSE TO REVIEWER 4 COMMENTS

Point 1:

L 27-58: Introduction part needs to add more recent reviews with significance. Author should add more details about study area such as Geographical coordinates, altitude, and temperature. There is no detailed methodology about data sources, collection and analysis, authors are requested to add in detail.

Response 1:

Introduction has been revised to include more recent reviews. Details around the data has been added, such as the National Land Cover Products (NLC products) from DEA. Geographical coordinates have now been added to describe the study area.

Point 2:

L 242-245: “Mining activities have contributed to economic development in many parts of the world (Clemens et al.2018). Mining projects contribute to employment opportunities and infrastructure development, such as roads and access to water ecosystems …”. It’s a general statement. Mention the kind of mining activities in the study area and discuss more with your results.

Response 2:

The paragraph has been revised to include the mining activities specific to the study area.

Point 3:

L 362: “Some of the drivers of change include agriculture, siltation, and concentration of chemicals….. which affects the world’s threatened flora, fauna, and organisms…”. Is it same drivers in your study area? Please discuss with your results.

Response 3:

Discussions using results of the study have now been added.

Point 4:

Table and Figures: Should correct the text formats uniformly (Eg: Legend; Axis Title)

Response 4:

Tables and figures have now been amended as suggested.

Point 5:

Add coordinates in all the figures

Response 5:

Coordinates have been added to Figure 1, 2 and 3

Point 6:

Whole manuscript needs to format according to the journal instructions. I recommend the manuscript for publication with “Minor Revision”

Response 6:

The authors attended to the suggested changes, and would like to thank the reviewer for pointing out the revisions. We also than the reviewer for recommending the publication of the manuscript. MDPI English editing services were used for language editing as well.

Round 2

Reviewer 1 Report

Thanks for extended introduction, now it is much better! Also description of the questioning and its addressing is clearer and better arranged. 

To point 3 -- map projection is OK, but figure 3 was probably resized without ratio and was flattened in previous version. Now it is OK 

To table 2 I can see that you have subtracted percentage of area in 1990 and 2018. I supposed something little different -- the ratio (percentage)  2018/1990. You can see, that e.g. water bodies were extended 25 times.. It is up to you if you want to add it. 

I do not have any other comments to other corrected parts, it is ok. 

Author Response

Please see attached response document.

Reviewer 2 Report

The authors have revised the manuscript considerablely. I have no further comment.

Author Response

Thank you.

Reviewer 3 Report

Line 68: Instead of starting the sentence with reference number (12), please start with the author name “Lawler et al. 2014 (12)”. Line 91: same comment. Please make sure that is not the case throughout the manuscript.

Figure 5 and 6 are exactly the same, I am not sure why two figures are required.

English is not my first language. I recommend the paper reviewed just for language, if possible.

Author Response

Attached, please find a document detailing the responses.
